# Diet and Nutrients in Rare Neurological Disorders: Biological, Biochemical, and Pathophysiological Evidence

**DOI:** 10.3390/nu16183114

**Published:** 2024-09-15

**Authors:** Marilena Briglia, Fabio Allia, Rosanna Avola, Cinzia Signorini, Venera Cardile, Giovanni Luca Romano, Giovanni Giurdanella, Roberta Malaguarnera, Maria Bellomo, Adriana Carol Eleonora Graziano

**Affiliations:** 1Department of Medicine and Surgery, “Kore” University of Enna, 94100 Enna, Italy; marilena.briglia@unikore.it (M.B.); fabio.allia@unikore.it (F.A.); rosanna.avola@unikore.it (R.A.); giovanniluca.romano@unikore.it (G.L.R.); roberta.malaguarnera@unikore.it (R.M.); maria.bellomo@unikore.it (M.B.); 2Department of Molecular and Developmental Medicine, University of Siena, 53100 Siena, Italy; cinzia.signorini@unisi.it; 3Department of Biomedical and Biotechnological Sciences, University of Catania, 95123 Catania, Italy; cardile@unict.it

**Keywords:** rare neurological disorders, nutritional compounds, dietary pattern, rare leukodystrophy, rare tumors, orphan disease, neurodegenerative disease

## Abstract

**Background/Objectives**: Rare diseases are a wide and heterogeneous group of multisystem life-threatening or chronically debilitating clinical conditions with reduced life expectancy and a relevant mortality rate in childhood. Some of these disorders have typical neurological symptoms, presenting from birth to adulthood. Dietary patterns and nutritional compounds play key roles in the onset and progression of neurological disorders, and the impact of alimentary needs must be enlightened especially in rare neurological diseases. This work aims to collect the *in vitro*, *in vivo*, and clinical evidence on the effects of diet and of nutrient intake on some rare neurological disorders, including some genetic diseases, and rare brain tumors. Herein, those aspects are critically linked to the genetic, biological, biochemical, and pathophysiological hallmarks typical of each disorder. **Methods**: By searching the major web-based databases (PubMed, Web of Science Core Collection, DynaMed, and Clinicaltrials.gov), we try to sum up and improve our understanding of the emerging role of nutrition as both first-line therapy and risk factors in rare neurological diseases. **Results**: In line with the increasing number of consensus opinions suggesting that nutrients should receive the same attention as pharmacological treatments, the results of this work pointed out that a standard dietary recommendation in a specific rare disease is often limited by the heterogeneity of occurrent genetic mutations and by the variability of pathophysiological manifestation. **Conclusions**: In conclusion, we hope that the knowledge gaps identified here may inspire further research for a better evaluation of molecular mechanisms and long-term effects.

## 1. Introduction

According to the World Health Organization (WHO), neurological disorders (NDs) are actually the leading cause of illness and disability worldwide [1,2]. A recent study shows that in the last decades, the number of people suffering from nervous system (NS) disorders (43% of the world’s population—3.4 billion people—affected in 2021) has sharply increased [3] and, even if the statistics report that tension-type headache and migraine represented the main disability-causing diseases, among the fatal conditions directly attributable to nervous system health loss, there were neonatal encephalopathy, Alzheimer’s disease and other dementias, meningitis, epilepsy, and nervous system cancer. The nervous system (NS) is vulnerable to various disorders and can be damaged by inflammatory processes or by immunological-mediated mechanisms, genetic disorders, traumatic injuries, and/or cancers. Moreover, neurological health loss is often a consequence of various circumstances, such as neonatal (premature birth, jaundice, and sepsis) and congenital conditions (birth defects and chromosomal abnormalities), infectious diseases (COVID-19, echinococcosis, cystic, syphilis, malaria, and Zika virus disease), or comorbidity with metabolic disorders (diabetic neuropathy) [4]. Although the etiology is heterogeneous, the NDs show systemic and long-term organic disabilities with a high incidence of death, as supported by meta-analysis studies [5]. In this context, one of the most ambitious purposes of global health systems is to improve the quality of life for people with NDs by reducing the influence of NDs, as well as their associated mortality, morbidity, and disability [6], as established by the World Health Assembly with the Global Intersectoral Action Plan on Epilepsy and Other Neurological Disorders 2022–2031 (IGAP). 

Although there is a high incidence of neurological disorders, unfortunately, to date, there are few effective cures. Despite the many advances made with the implementation of therapeutic solutions, there are problems and limitations that do not guarantee full control of neurological symptoms [7], suggesting that dietary nutrition may represent a useful tool. The dietary pattern and nutritional compounds play a key role in ensuring general physical wellness and specific neurological function. Daily dietary intake provides nutrients and molecules to support life and health maintenance through its general supply of substrates of homeostasis. Dysphagia, movement disorders, cognitive impairment, and depression associated with neurological disorders can directly or indirectly affect the nutritional status of patients. For instance, malnutrition causes delays in rehabilitation and induces an increase in both mortality and morbidity. Proper nutritional education could be a crucial intervention in ND treatment to minimize the risks of malnutrition with likely repercussions on the worsening of symptoms related to neurological diseases [8].

In recent years, epidemiological data have revealed the importance of a correlation between both healthy food intake and lifestyle and a clear reduction in the risk of central nervous system (CNS) diseases. These findings make diet and lifestyle interesting points on which we can focus the intervention research field [9,10]. Particularly, it has been scientifically proven that the intake of certain foods and nutrients guarantees clinical benefits for CNS disease. Currently, research on brain health and the effects of nutraceuticals receives high interest due to the potential neuroprotective effects.

In this work, we focus our attention on the effects of diet and/or nutrient intake on some rare diseases with neurological symptoms. The term “Rare diseases” (RDs) refers to a wide and heterogeneous group of multisystem life-threatening or chronically debilitating clinical conditions. They often correlate with high medical costs, poor quality of life, reduced life expectancy, and a significant mortality rate in childhood. Examples include genetic diseases, rare cancers, and infectious tropical diseases. 

The only finding that RDs have in common is their low prevalence and low frequency in the population. According to the definition by the WHO, an RD is an illness or condition that occurs from 0.65 to 1 case per 1000 population, with a prevalence from 6.5 to 10 cases per 10.000 residents. Thus, their classification criterion is usually purely epidemiological. Thus, even if many pathologies reach just a prevalence of 0.001%, (1 case per 100,000), taken together, RDs affect approximately 6–8% of the world’s population, and often, the rarity of a particular disease limits the process of drug development for economic factors [11]. 

In this background, we hypothesized that basic research, clinical practice, and evidence-based medicine could furnish us with the data to investigate new therapeutic perspectives by establishing a direct link between diet and neurological outcomes in a specific “rare” molecular context. Therefore, in this narrative review, we first introduce readers to the topic of rare diseases by focusing on rare neurological disorders. Then, we analyzed the concept of “nutritional approaches” in neurological illness, trying to summarize the main methodologies applied for the study of nutritional interventions in rare neurological disorders with a critical evaluation of the limits and strengths of *in vitro* and *in vivo* models and of clinical studies. These preliminary indications supported the last part of the work in which we analyzed the scientific literature by reviewing the *in vitro*, *in vivo*, and clinical evidence on the effects of diet and of nutrient intake on some rare neurological disorders, including rare developmental disorders (Angelman Syndrome and Rett Syndrome), rare leukodystrophies (Krabbe disease and Pelizaeus–Merzbacher disease), rare genetic epilepsy, rare forms of ataxia, and rare brain tumors. Specifically, we review the evidence available on the following web-based databases: PubMed, Web of Science Core Collection, and DynaMed. Moreover, for clinical trials, we searched on clinicaltrials.gov. Globally, to focus our search, we applied the Boolean operator “AND” to combine the condition/disease with specific terms (diet, or nutrition, or nutrients). When a specific term turned results in line with the aims of this review, closer research was performed by the inclusion of the identified key work.

We try to sum up and improve our understanding of the emerging role of nutrition as both first-line therapy and risk factors in rare neurological diseases, in line with the increasing number of consensus opinions suggesting that nutrients should receive the same attention as pharmacological treatments. The limits, the challenges, and the knowledge gaps are identified in order to inspire further research for a better evaluation in the future.

## 2. Rare Disorders: Lights and Shadows

The term “Rare diseases” (RDs) refers to a narrow subset of diseases affecting a small group in the general population with a wide spectrum of clinical conditions. The findings shared by all the RDs are the low prevalence and low frequency in the population; they are orphan diseases (neglected diseases) [12]. Unfortunately, only a small group of these diseases can be predicted in possible treatment, in terms of relieving the symptoms and improving the quality of life [13]. From a medical perspective, rare neurological disorders include many rather complex and heterogeneous conditions, for which knowledge of natural history, diagnosis, prevention, neurobiology, and progress in treatment/drug strategies is almost entirely limited [14]. The designation is linked to disease prevalence and severity and the existence of alternative therapeutic options, but it varies across jurisdictions. A disease is listed as rare when its prevalence conventionally does not exceed a certain threshold. This latter varies among countries that adopt slightly different parameters. In the European Union (EU), this threshold is fixed at 0.05% of the population, (1 in 2000 people or 5 per 10,000). In the United States (US), a disease is considered rare when it does not exceed the prevalence of a 0.08% threshold and consequently, it affects fewer than 200,000 patients in the country (6.4 in 10,000 people); whereas, in Japan, a disease is rare if it includes less than 50,000 cases (4 cases per 10,000). Despite the low prevalence, the number of rare diseases for which no treatment is available is estimated to be between 6000 and 8000 worldwide. It is worrying that approximately 6% of the world’s population is affected by potentially harmful and/or lethal rare diseases; and it is expected that the value could grow with the progress of science and, in particular, with advances in genetic research. Especially in conditions with a genetic etiology, next-generation sequencing techniques and the sequencing of the whole exome could play a key role [15,16]. Data show that a high prevalence of rare diseases has a genetic origin and mainly involves children [17]. It would seem, at least in part, that environmental exposure during pregnancy or in the early stages of life may affect genetic vulnerability. The main cause lies in rare tumors, autoimmune diseases, toxic and infectious diseases, or congenital malformations [18].

RDs constitute a priority public health topic by the European Commission and the agencies for public health due to their high number and their complex management that involves not only medical but also social issues [19]. In fact, RDs share a number of common features related to social and health burdens. Among these, the poor awareness among healthcare professionals and often the inadequate levels of care knowledge are consequences of their low prevalence. Often, all these findings turn into delayed diagnosis, misdiagnosis, or even un-diagnosis and troubles in disease management. Scientific evidence indicates that the delay in diagnosis is due to the lack of knowledge about the topic and to the gradual manifestation over time of clinical characteristics. In many cases, genetic testing would be necessary [20,21]. Moreover, the restricted market addressed to the single conditions reduces the attractivity of the pharmaceutical industry to invest in Research and Development (R&D) for new drugs and treatments that have huge development costs, resulting perhaps in the most expensive produced drugs [22]. For all these reasons, extraordinary support in providing high-quality information on rare diseases by increasing knowledge and ensuring improvement in healthcare and research is carried out by several international health committees. For example, the Orphanet provides many data and specialist services around the world through the ORPHA code that classifies rare diseases [23]. Moreover, the European Reference Network for Rare Neurological Diseases [24], the European Reference Network [25], and the ITHACA network (on rare congenital malformations and rare intellectual disabilities) [26] provide excellent support to the cooperation of researchers and health professionals in this field. 

### Rare Neurological Disorders

Rare neurological diseases could have an hereditary, post-infectious, iatrogenic, or unknown etiology. These pathological conditions cause damage to the brain, spinal cord, or peripheral nerves. Since the nervous system controls more functions of the body, depending on the pathology, the symptoms of a rare neurological disease can be many, and they range from mild tremors to severe motor and cognitive damage. Usually, it has been commonly described as the progressive cognitive decline toward dementia associated with personality changes and psychiatric disorders, the loss of coordination, mobility, muscle strength, and balance with tremors and dystonia, as well as vision and hearing problems and involuntary movements [27].

According to the Orphanet, the term “Rare neurological disorders” does not characterize a disease but a group of diseases with genetic causes (ORPHA:71859). The number of rare neurological disorders that are also orphan diseases is about 8000, a number which tends to increase monthly with advances in genetic science and research [28]. The identification of rare disorders as well as the rare expression of common disorders associated with these pathological conditions is particularly difficult, even if technological, financial, and social changes have taken place in recent decades [29]. To better characterize patient populations and delimit target populations, natural history studies are highly recommended. These latter are epidemiological studies focusing on frequency description and the characteristics and evolution of disease by collecting real data from groups of patients affected by these diseases. These studies are often performed early in the clinical development process to support and guide the design of clinical trials and drug development [29,30,31].

Therapy is often supportive. In recent years, pre-clinical and/or clinical studies have addressed more attention toward personalized medicine and global health with specific attention dedicated to nutritional interventions as both first-line therapy and risk factors, suggesting that nutrients and alimentary wellness should receive the same attention as pharmacological treatments [32,33,34]. Figure 1 reports the main characteristics of rare neurological diseases and possible investigations for symptomatic treatments.

## 3. The Alimentary Wellness for a Global Health 

A varied and balanced diet is the basis of a healthy life. On the other hand, an unbalanced diet affects psycho-physical human health and represents one of the main risk factors for the onset of chronic diseases. In the last two decades, lifestyle and wrong eating habits have heavily affected the increased prevalence of metabolic diseases such as diabetes, obesity, cardiovascular disease, and fatty liver [35,36,37,38]. To address this growing public problem, health organizations have provided recommendations on proper nutrition intake [39]. 

Recent advances in research have improved the understanding of metabolism and highlighted the active involvement of nutrients and their metabolites in the regulation of gene expression and cell functions. 

The assimilation and transformation of food including the action of nutrients and non-nutritive components are closely related to extrinsic factors (e.g., food, xenobiotics, and environment), intrinsic factors (e.g., gender, age, and gene changes), and the host/microbiota interaction (Figure 2) [33,40,41]. The aforementioned factors influence nutrient metabolism as well as the risk of developing various metabolic diseases. Extrinsic factors play an important role in the metabolic activity of nutrients and in ensuring a healthy condition.

For instance, physical factors such as photoperiod and temperature are involved in the regulation of metabolic activity. This latter is influenced by circadian endogenous rhythms [42]. Exposure to harmful environmental conditions induces the release of stress hormones, which can impair the body’s ability to perceive and respond to metabolic changes [43,44,45]. Moreover, extrinsic factors can stimulate the alterations of the epigenome with consequent lasting effects on the energy and nutrient metabolism, determining the development of metabolic disorders such as in the heart (coronary heart disease) and in the brain (for example, Alzheimer’s disease) [46,47,48]. The intrinsic factors, including gene changes, gender, and age, influence the proper functioning of metabolic pathways [49].

It is clear that the modulation of central nervous system homeostasis by nutrients could be both a risk factor for the development of neurological disease [50] as well as an opportunity to improve the health status by nutritional interventions. 

### 3.1. Nutritional Interventions: What Are They?

Nutritional interventions were defined as “purposefully planned actions intended to positively change a nutrition-related behavior, environmental condition, or aspect of health status” [51,52]. Any specific strategy designed to improve the health status of a patient by modifying its dietary habits is a nutritional intervention. The suggestion of specific nutrient intake or the consumption of a particular food might bring some benefits regarding CNS health. In clinical terms, nutritional approaches can be categorized into two major types: nutrient supplementation and dietary modifications.

Some clinical benefits on metabolic health, neural function, and longevity seem to be related to nutritional interventions in cases of neurological conditions with high healthcare and social costs.

In this section, a brief overview of nutritional approaches for the most common neurological diseases was conducted in order to better compare this scenario with that of rare neurological disorders.

#### 3.1.1. The Nutrient Supplementation in Neurological Illness

In the case of nutrient supplementation, the intake of specific nutrients by dietary supplement is conducted from different sources (plant extracts, prebiotics, vitamins, amino acids, fibers, metals, fatty acids, etc.) as complementary and integrative treatments. Considering that the pathophysiology of neurological disorders is related to oxidative stress, neuroinflammation, and mitochondrial dysfunction [53], it has been shown that the action of some dietary factors on mitochondrial dysfunction, epigenetic modification, and neuroinflammation are mechanisms that would seem to underlie the action of nutrients on brain health [54]. 

Micronutrients (vitamins and trace elements) are essential components for metabolic processes. Micronutrients play an important role through catalytic action in enzyme systems like cofactors and components of metalloenzymes. They are also involved in antioxidant activity, modulation of cellular immunity, and wound healing [55]. Acute and chronic changes in micronutrient levels can cause complications in neurological diseases [56]. Actually, micronutrient dysregulation can cause damage to peripheral nerves (demyelination or axonal damage), impairment to the central nervous system, and a typical category of myeloneuropathy (damage to the peripheral and central nervous system) [32]. Globally, vitamins and inorganic ions seem to be useful—as complementary nutrients—for the prevention and management of neurological diseases [56].

Several pieces of scientific evidence show encouraging results on the synthesis of neurotrophic factors and neurotransmitters, neuroplasticity, myelination, and microglial activity as nutritional effects related to the intake of vitamins and minerals [57]. Specifically, as for vitamins, it has been reported that the vitamins of the B group (B6 pyridoxine, B9 folic acid, and B12 cobalamin) may slow the progression of cognitive decline in patients with mild cognitive impairment [58,59] by a reduction in homocysteine levels. A slower progression of Alzheimer’s disease was reported in patients at disease onset when supplemented with vitamin E, which, as an antioxidant, protects neurons from oxidative damage [60]. Supplementation with vitamin D has been reported to reduce the frequency of relapses and to slow disease progression in subjects affected by Multiple Sclerosis [61,62].

Among the inorganic elements of bio-importance to health, magnesium and iron have been largely evaluated in neurological dysfunctions [63], and their supplementation seemed to support neurological function [64,65,66].

Dietary intake of fish and omega-3 fatty acids has been associated with a lower risk of Alzheimer’s disease [67,68].

A wide variety of natural plant substances are known as “neuro-nutraceutical” substances [34]. *In vitro* studies showed neuroprotection effects given by seed extracts as a result of their antioxidant, anti-inflammatory, and anti-apoptotic properties [69]. Potential beneficial effects are related to several cannabinoid compounds extracted from *Cannabis sativa*. For instance, cannabisin F in a model of inflammation and oxidative stress induced by lipopolysaccharides in BV2 microglial cells showed a significant reduction in both inflammatory responses and production of reactive oxygen species associated with the expression pathway of the enzyme sirtuin 1/nuclear factor kappa B and the nuclear factor 2 related to erythroid-2. Moreover, the antioxidant activity of *Cannabis sativa* seed extract plays a role in the reduction of reactive oxygen species and in the expression of nuclear factor 2 related to erythroid-2 and oxygenase-1 of heme (HO-1) [70,71]. Bhuiyan et al. demonstrated the neuroprotective effect of anthocyanins extracted from black soybean seed coat using a model of ischemia induced by oxygen–glucose deprivation and cell death induced by glutamate in primary cortical neurons of rats [72].

Globally, we want to highlight that the major limit seemed to be that often, in the analyzed data, the baseline of nutrient levels was not fully reported at a specific stage of disease progression. So, the supplementation can have an individual effectiveness according to the severity of the specific neurological condition. Moreover, the epidemiological analysis of the possible correlation between nutrient consumption and cognitive decline is complex and laborious. It is unlikely that a single component, alone, plays an important role considering that many factors throughout life affect brain function. Therefore, multi-domain interventions could be more promising in the attempt to prevent cognitive decline. Nowadays, designing this type of trial is challenging for researchers [32].

#### 3.1.2. The Dietary Modification in Neurological Illness

Dietary modification involves the change in the types or quantities of consumed food. Often, it is a diet restriction (DR) of particular nutrients (carbohydrates, amino acids, etc.) or a time-limited diet, such as intermittent fasting (IF) or a fasting-mimicking diet [7].

According to the literature analysis, in order to manage the most common neurological illnesses and potentially delay their progression, the principal dietary modifications are the Mediterranean Diet, the Ketogenic Diet, the Paleo diet, and the Low-Fat or the Low-Protein Diets [7,73,74].

Globally, the major reported limit is the adherence to a specific diet. Moreover, dietary preferences can be conditioned by single nucleotide polymorphisms in genes coding for taste receptors [75].

The Mediterranean Diet is characterized by a high intake of fruits, vegetables, monounsaturated fat, fish, whole grains, legumes, and nuts. Alcohol consumption is moderate, as well as the consumption of red meat, saturated fat, and refined grains. At the nutritional level, the contents of omega-3 unsaturated fatty acids, resveratrol, and micronutrients are high. It has been associated with a reduced risk of cognitive decline, Alzheimer’s disease [76,77,78], and Parkinson’s disease [79,80,81]. In the case of Multiple Sclerosis, the Mediterranean diet has been reported to be effective for prevention and for a reduction in comorbid disease severity, but the reduction in symptoms has been related to different diets such as the low saturated fat (Swank), low fat vegan (McDougall), modified Paleolithic (Wahls), and gluten-free diets as well as intermittent fasting, calorie restriction, and intermittent calorie restriction (fasting mimicking diet) [82].

The ketogenic diet is a high-fat low-carbohydrate diet that induces ketosis. It seems to improve cognitive function in Parkinson’s disease [83,84]. It seems that the produced ketones may serve as an alternative energy source for the brain, potentially reducing the impact of glucose hypometabolism in Alzheimer’s patients [85]. A lot of papers supported the role of the ketogenic diet in the treatment of epilepsy [86,87,88,89], probably associated with the stabilization of neuronal activity and reduction in seizure frequency by shifting the body’s metabolism from glucose to ketone.

Early dietary interventions in children with Down syndrome have been reported as an opportunity for decreasing the risk or delaying some symptoms, ameliorating their quality of life [90].

## 4. Nutritional Interventions for Rare Neurological Disorders: What Is the Right Model?

Studying rare diseases is challenging, primarily because of their different etiology, the huge variety of symptoms, the variations in the time of onset, and the low prevalence in the total population.

This complexity necessitates innovative approaches and *ad hoc* models applicable both to better clarify the disease mechanisms and to screen and develop potential treatments or to evaluate supporting nutritional care.

In preclinical studies, *in vitro* models are useful tools for basic research and therapeutic development, while *in vivo* animal models are valuable resources to confirm *in vitro* speculations, especially when investigation in human subjects may be limited by ethical concerns or the limited availability of patients. In order to highlight the importance of tailored research methodologies, we analyzed the literature background to define how these diseases are studied/treated and to evaluate how preclinical research could mirror the genetic context of the patient with specific regard toward nutritional interventions.

### 4.1. In Vitro Models for Testing Nutritional Interventions in Rare Neurological Disorders

Concerning neurological diseases, the effects of nutrition and the mechanisms involved are still unclear. As it is still hard to understand the mechanism of rare neurological diseases on the basis of only clinical patients or animal models; cell models cultured *in vitro* play a key role, representing excellent tools in the research for disease-causing mechanisms and in the therapeutic development for screening and testing potential treatments.

The common and most long-lasting used *in vitro* model is the two-dimensional (2D) culture system. According to the examined disorder, it consists of the selection of the appropriate cell type (or types) that is cultured *in vitro* and exposed to the neurotoxic stimulus, a hallmark of each disease. In this context, the effects of nutrients could be investigated. This cell model allows the use of human cells, unraveling the doubts due to species differences. The advantages of using *in vitro* models are the reduced costs, the ability to study specific biological endpoints, and the medium and high throughput analysis of the experiments [91], but they fail in the reproduction of the microenvironment of human tissues. Major progress has been made in this field by the emerging three-dimensional (3D) culture [92] and induced pluripotent stem cell (iPSC) technologies.

Compared to 2D culture systems, in 3D culture systems, coherent cells within an extracellular matrix better reflect physiological behavior as in a real tissue environment, enabling the study of the interaction of different cell types [93]. In particular, iPSCs derived from the patient can be used to create patient-specific cell lines. These cell lines accurately reflect the disease phenotype at the cellular level, allowing a relatively accurate assessment of pharmacological responses and toxicity in the patient’s genetic field [94,95]. This application strategy not only improves the relevance of preclinical results but may also lead to personalized therapeutic approaches.

In recent years, the major innovation for both 2D and 3D *in vitro* models is the application of High Throughput Screening (HTS) technology [96,97,98,99]. It consists of a miniaturized assay and large-scale data analysis to rapidly identify active compounds, antibodies, or genes modulating the molecular pathways involved.

At a glance, the major remarks are reported in Figure 3. Although *in vitro* studies ensure the possibility of prediction of clinical outcomes, they are limited as they cannot reflect the physiological processes of absorption, distribution, metabolism, and excretion.

### 4.2. In Vivo Models for Testing Nutritional Interventions in Rare Neurological Disorders

Several model organisms have been chosen to better understand the characteristics of rare neurological disorders and identify causative genes. Moreover, with respect to *in vitro* models, the *in vivo* models admit the evaluation of nervous functions, such as motor function, learning and memory, and physiological development. In this context, the effects of nutrients, diet, and nutraceuticals should better mimic the clinical outcomes.

As reported in Figure 4, the classical model organisms for *in vivo* studies are mammalian, especially mice; however, studies on non-mammalian model organisms are becoming increasingly common. Fruit flies (*Drosophila melanogaster*), nematode worms (*Caenorhabditis elegans*), and zebrafish (*Danio rerio*) mainly provide advantages in terms of a rapid and economic evaluation of the effects of nutrition correlation with gene variants. Subsequently, data obtained can be validated in mammalian model organisms, such as mice, and in human cells [100].

Functional studies in model organisms with informatics support can also be used to make diagnoses. The Model Organism Screening Center uses model organisms to provide evidence of the pathogenicity of genetic variants identified in Undiagnosed Diseases Network patients. Worms (*Caenorhabditis elegans*) and zebrafish (*Danio rerio*) are considered excellent models due to numerous advantages including high gene homology with humans, low maintenance costs, and rapid development. In addition, transparency—in certain stages—allows the detection of cellular and morphological defects [58].

Of course, when the disease-causing agent is a genetic mutation, gene knockout/knockin, chimeric, or chemically-induced organism models have been developed in all taxa [101].

#### 4.2.1. Nonhuman Primate Genetic Models

As recently reviewed by Vallender and co-workers, naturally occurring nonhuman primate models (macaques) of rare human neurological diseases are being discovered and developed [102]. This is the case for late infantile neuronal ceroid lipofuscinosis, Krabbe disease, Leukoencephalopathy with Ataxia, and Pelizaeus–Merzbacher disease. Even if the macaques physiology is closely similar to the human one, the application of primate models—especially for nutritional intervention—is limited as a consequence of their cost and ethical affairs [103].

#### 4.2.2. Rodents

Rodents (mice, rats, etc.) have been largely applied for studying rare neurological disorders. Their application is sustained by some advances: (i) small size, (ii) short generation times, (iii) short life cycle, (iv) easy maintaining and breeding, (v) similarities to humans in terms of anatomy and physiology, and (vi) possibility of genetic manipulation [104,105]. Meanwhile, some advantages could also represent limits, for example, the small size is linked with difficulties in experimental procedures and differences in metabolism [106]. A naturally occurring mouse model has been reported for some rare disorders like Krabbe disease [107], but knockout mouse models seemed to be the major resource for the study of rare diseases [108].

A system of synergistic data-sharing has been developed to limit the number of animals and to ensure the opportunity to discover still unknown therapeutic approaches (for example, dietary intervention or drug therapy) useful for patients [109,110].

#### 4.2.3. *Caenorhabditis elegans* (*C. elegans*)

The microscopic nematode worm *Caenorhabditis elegans* is a model organism suitable for rare neurological disease modeling [111,112]. The whole genome of this organism has been sequenced and it is the only organism to have completed its connectome (the “wiring scheme” of neurons). It has also been shown that 83% of the proteome is characterized by human homologous genes [113]. These data show that *C. elegans* is suitable for functional human gene research, especially for studies on neural mechanisms and molecular learning, memory, coupling behavior, chemotaxis, thermotaxis, and mechanical transduction [114]. Recent advances in genetic manipulation have facilitated the precise manipulation of genes at a single nucleotide level using the CRISPR-Cas9 gene editing technology [115,116]. In addition, investigative studies on *C. elegans* are supported by well-established knowledge, which are valuable publicly accessible resources (e.g., WormBase, https://wormbase.org//#012-34-5; Caenorhabditis Genetics Center https:/cgc.umn.edu) to ensure the integration of new findings on disease pathologies.

To date, the *C. elegans* model aided the characterization of a novel gain-of-function mutation in Sodium Leak Channel Non-selective (NALCN) identified in a girl with intellectual disability, episodic and persistent ataxia, and arthrogryposis [117]. Moreover, genetic defects associated with rare ciliopathy disorders (Joubert syndrome, Meckel syndrome, and nephronophthisis) have been modeled in *C. elegans*, showing severely disrupted ciliary function and structure [118,119].

A nutritional intervention was tested in a *C. elegans* model for maple syrup urine disease, a rare genetic disorder of aminoacidic metabolism characterized by a deficiency of an enzyme complex (branched-chain alpha-keto acid dehydrogenase) that is required to break down the branched-chain amino acids (BCAAs) [120].

However, it should be noted that the application of *C. elegans* in an *in vivo* nutrigenomic model is limited by the high-protein low-fat low-carbohydrate diet, as it naturally feeds on living bacteria.

However, it should be noted that the application of *C. elegans* as an *in vivo* nutrigenomic model is limited by the high-protein low-fat low-carbohydrate diet, as it naturally feeds on live bacteria.

#### 4.2.4. *Drosophila melanogaster*

The fruit fly *Drosophila melanogaster* is a very attractive *in vivo* model for basic research and large-scale screening experiments [121], as a consequence of the high progeny production, and the short time generation for each reproduction cycle. The genome has been fully sequenced [122] and it shows excellent preservation between human and fly genomes (65% for coding genes and 80% for human genes associated with disease) [123,124]. In addition, highly sophisticated genetic tools are available to manipulate the genes of interest in a controlled manner, making investigational studies on rare human variants with unknown pathogenicity in an appropriate cellular or molecular enviroment [125,126,127].

Interestingly, *Drosophila* has been reported as a diet discovery tool for treating rare disorders of amino acid metabolism disorders, offering the opportunity to generate and test the disease-relevant phenotypes and to perform high-throughput targeted diet screening [128,129].

#### 4.2.5. Zebrafish (*Danio rerio*)

The zebrafish (*Danio Rerio*) is a small tropical freshwater fish easy to maintain in an animal facility at low costs. It is a valid model organism also for the transparency of embryos, the rapid development, and the opportunity to make real-time imaging of cells and internal structures under a microscope. In addition, the embryos develop externally and a pair of fish is able to lay several hundred embryos in a single brood [130]. It has been demonstrated that 82% of known human genes related to the disease and 76% of human genes involved in genomic association studies have orthologues in zebrafish [131]. With the impactful developments of CRISPR and next-generation sequencing technology, zebrafish models have gained increasing success and high approval in biomedical research [132], especially for rare neurological disorders [133,134].

As for the evaluation of dietary intervention in the zebrafish model of rare disease, the assessment of supplementation with cobalamin in the rare combined methylmalonic aciduria and homocystinuria has been performed by adding supplements in water [135]. Globally, the major obstacle is the solubility of some nutrients that could leach into the tank water.

### 4.3. Clinical Research for Testing Nutritional Interventions in Rare Neurological Disorders

The power of clinical studies is often limited by the small participant pool that is often restricted by rigid inclusion and exclusion criteria. In addition, the severity of symptoms often varies over time and test results only reflect the patient’s condition at a specific time [136]. These pieces of evidence suggest that it is necessary (i) to conduct global multi-center recruitment; (ii) to improve the effectiveness of study projects with the cooperation of doctors, pharmaceutical companies, government agencies, clinical research methodologies, biologists, patients, and their families [137]; and (iii) to accelerate the development and application of shared and objective outcome measurements by biomarkers identification. To overcome these limitations, the National Institutes of Health (NIH) supported the Clinical Research Network on Rare Diseases (RDCRN), which was established to promote the diagnosis, management, and treatment of rare diseases and to ensure highly effective and efficient collaboration multi-site, patient-centered, pro-translational, and clinical research. It also works in close contact with patient advocacy groups. Up to now, the RDCRN has published several articles on topics ranging from natural history study results and case reports to practical guidelines and clinical trials. In addition, the RDCRN has been involved in work, which, as a result, has produced 10 treatments approved by the Food and Drug Administration (FDA) including the approval of trofinetide as the first treatment for Rett syndrome [138,139,140,141,142]. An increasing number of technologies are now being used for remote health monitoring by reducing cohort sizes and endpoint response times for clinical trials and by having a clear picture of the patient’s status [143]. Digital measures have mainly been developed for monitoring neurological disorders that are not classified as rare [144,145,146], but these measures are increasingly being used for evaluations in patients with rare diseases. The use of digital measures of results is expected to increase with the development of innovative technologies and artificial intelligence [147].

As recently reported by Liu et al. [148], the research on diet therapy in patients with rare diseases focuses on diet therapy methods, diet therapy management, guidelines for diet therapy, and the impact of diet therapy on patients. They are mostly related to rare inborn errors of metabolism [149].

## 5. Diet and Nutrient Intake in Rare Neurological Disorders

### 5.1. Rare Pervasive Developmental Disorder

#### 5.1.1. Angelman Syndrome

The clinical manifestations of Angelman syndrome include mental retardation, movement or balance disorder, typical abnormal behaviors, and severe limitations in speech and language. The etiology resides primarily in *de novo* maternal deletions involving chromosome 15q11.2-q13; but it is also associated with paternal uniparental disomy for the same chromosome, imprinting defects, or mutations in the gene encoding the ubiquitin-protein ligase E3A [150]. As for diet and nutrient intake in Angelman Syndrome, we found 2 clinical trials and 19 articles. Among the latter, only 13 dealt with nutritional interventions. Considering that Angelman syndrome is often associated with refractory epilepsy [151], the ketogenic diet [152,153] and the low glycemic index diet [154,155] have been suggested in clinical practice.

In mouse models of disease, (i) the ketone ester (R,S-1,3-butanediol acetoacetate diester) supplementation attenuated seizure activity and enhanced synaptic plasticity, improving the motor coordination, learning, and memory [156] and (ii) the supplementation with linoleic acid ameliorated the mechano-sensory deficits by acting on increasing the reduced activity of the PIEZO2 ion channel [157].

These results strongly seemed to support a clinical trial (ClinicalTrials.gov ID NCT03644693) that proved the safety and tolerability of a nutritional formulation containing exogenous ketones [158,159]. Moreover, dietary supplementation with methylation-promoting agents (betaine, metafolin, creatine, and vitamin B 12) was evaluated in a clinical trial (ClinicalTrials.gov ID NCT00348933) but resulted in being ineffective in decreasing the severity of the disease [160,161].

#### 5.1.2. Rett Syndrome

Rett syndrome is a neurodevelopmental disorder first observed by the Austrian pediatrician Andreas Rett in two girls having typical hand-wringing stereotypes. It affects mainly, but not exclusively [162], female subjects [163].

The progression of pathology has been fully characterized [164,165,166,167]. Briefly, normal growth up to the first 6 months of life is followed by failure to reach the physiological developmental stages between 6 and 18 months. At 12–30 months, a period of regression appears with gait dysfunction, loss of acquired hand skills, and spoken language, and the onset of repetitive hand stereotypies. From approximately 5 years of age through adulthood, the pseudo-stationary stage appears with no continued skill regression, with the exception of some loss of ambulation in the teen years. At last, the late motor deterioration stage can extend for years or decades. Common features include scoliosis, decreased mobility, muscle weakness, spasticity, or stiffness. Sometimes walking stops. The reduction in communication skills also triggers the appearance of autistic traits. Other severe clinical defects have been described such as apnea, hyperventilation, scoliosis, weight loss, and cardiac abnormalities in affected girls [168].

Rett syndrome (OMIM #312750) is primarily caused by loss-of-function mutations in the methyl-CpG-binding protein 2 (*MECP2*) gene located on the X chromosome (Xq28) [169]. Over 500 different MECP2 mutations have been identified. As reviewed by Good et al., there are missense, nonsense, frameshift, splice site, and start codon mutations as well as larger deletions [170]. A great effort has been made to evaluate the genotype–phenotype relationship [171] and genetic therapies [172]. Despite a wide phenotypic variability, RTT is commonly associated with epilepsy, sleep disturbances, and gastrointestinal dysfunction.

MECP2 encodes for the methyl-CpG-binding protein 2 (MeCP2), a DNA-binding protein with acts as a transcriptional modulator and epigenetic regulator of gene expression [169] (Figure 5). The protein is strongly expressed in the CNS on a time course that correlates with neuronal maturation and synaptogenesis [173]. Thus, it plays a critical role in brain development, particularly in regulating the expression of other genes important for synaptic function [174,175], as well as in chromatin organization, alternative splicing, and miRNA processing [176]. Moreover, mutations in other genes can also result in Rett-like syndromes, such as the X-linked Cyclin-Dependent Kinase-Like 5 (CDKL5; OMIM #300203) [177] or the Forkhead box G1 (FOXG1; OMIM #164874) [178] genes.

As for the cures, Trofinetide [179] has been recently approved for Rett syndrome [180]; and there are four recruiting clinical trials for gene therapy (ClinicalTrials.gov IDs: NCT05740761, NCT05898620, NCT05606614, and NCT06152237). Ongoing treatments are meant to alleviate disease symptoms, such as antiepileptic drugs, occupational and physical therapy, and scoliosis equipment.

Nutritional interventions (Table 1) for Rett disease have been suggested. Patients are characterized by weight deficiencies and poor growth, and are at risk of food shortages [181,182], vitamin D deficiency [183], osteopenia [184], and low bone mineral mass [185]. Moreover, changes in gut microbiota have been reported both in humans [186] and in MeCP2-deficient rats [187]. Recently, it has been reported that probiotic supplementation ameliorates neurological outcomes in Rett syndrome [188].

Motor and cognitive impairments may be related to cholinergic hypofunction abnormalities, and choline supplementation was found to alleviate the synaptic defects in iPSC-derived neurons [189] and to improve locomotor function in mouse models [190,191].

By searching for clinical trials with dietary intervention in Rett syndrome, we found two completed trials without published results: the NCT05352373 on the role of dietary calcium and the NCT 05420805 on pre- and post-biotics strategies. The creatine monohydrate supplementation was tested on Rett patients as a source of labile methyl groups for different methylation reactions (NCT01147575), but even if the DNA methylation increased, no significant differences were found in clinical parameters with respect to placebo [192].

From a biological and biochemical point of view, in zebrafish and in mutant mouse models as well as in humans, the major protein expression changes point out defects in energy metabolism, mitochondrial function, redox status imbalance, and muscle function [193,194], with annexed inflammation and oxidative stress [195,196]. In this context, the alteration in cholesterol and lipid metabolism [197] with a significant increase in lipid peroxidation [198,199] has been reported, and the diet supplementation with the ω-3 polyunsaturated fatty acids was found to improve the patient’s subclinical inflammatory status, partially restoring membrane fatty acids and correcting the redox status [200,201]. Moreover, an anaplerotic triheptanoin diet—concerning the stimulation of mitochondrial function—revealed a significantly increased longevity and improved motor function and social interaction in KO mice [202].

As reviewed by Mouro et al. [203], most studies focused on the control of epilepsy by supplementation with derivatives of the cannabis plant or by diet modification. Concerning cannabinoids, Cannabidivarin seemed to be useful to rescue cognitive deficits and to delay neurological and motor defects in MeCP2-mutant mice [204,205]. Due to the pandemic period and the recruitment challenges, a long-term safety study of cannabidiol oral solution in patients with Rett syndrome was terminated (NCT0425286), and the posted results report a reduction in seizure frequency in 43% of patients, while 5% became seizure-free. However, patients suffered from serious adverse side effects such as diarrhea, vomiting, fatigue, pyrexia, and somnolence.

As for diet modification, the ketogenic diet has shown good results on seizure frequency and behavior, as well as in the case of refractory epilepsy [206,207,208]. The antiepileptic actions seem due to the GABAergic signaling mechanism [209], or to an increase in adenosine and BDNF signaling [210].

**Table 1 nutrients-16-03114-t001:** Nutritional interventions for Rett syndrome.

Nutritional Interventions for Rett Syndrome		
	**Nutrients or Neuro-Nutraceutical Substance**	**Effects**
**Nutrient supplementation**	Vitamin D ^1^	Reduction in vitamin D deficiency [183]
	ω-3 polyunsaturated fatty acids ^1^	Improvement in inflammatory status [200,201]
	Triheptanoin ^2,3^	Amelioration of mitochondrial function, motor function and social interaction [202]
	Choline ^2^	Modulation of neuronal plasticity, possibly leading to behavioral changes.
	Probiotic ^1^	Amelioration of neurological outcomes [188]
	Cannabidivarin ^2^	Rescue cognitive deficits and delay of neurological and motor defects [204,205]
	Cannabidiol ^1^	Antiepileptic actions
	Creatine monohydrate ^1^	Increase in DNA methylation [192]
	**Type of diet**	**Effects**
**Dietary modification**	Ketogenic diet ^1,2^	Antiepileptic actions [206,207,208]

^1^ Tested in patients. ^2^ Tested in animal models. ^3^ Tested in *in vitro* model.

### 5.2. Rare Leukodystrophies

Although there are different causes of leukodystrophies, the patients share the manifestations of neurological symptoms. Among these are muscular spasticity, ataxia, seizures, cognitive developmental delay, dystonia, and dyskinesias. Moreover, swallowing dysfunction and pulmonary problems result in feeding limitations.

#### 5.2.1. Krabbe Disease

Krabbe disease (OMIM #245200) is one of the classic genetic lysosomal storage diseases with autosomal recessive inheritance. It is characterized by demyelination in the white matter of the central and peripheral nervous system. Clinical manifestations report infantile and late-onset forms. This pathology is due to mutations in the *galc* gene (Ch. 14q31), which encodes for galactocerebrosidase, the lysosomal enzyme that catalyzes the hydrolysis of galactose from galactocerebrosides and galactosyl-sphingosine (psychosine). The enzyme loses its function and, according to the “psychosine hypothesis” [211,212], the accumulation psychosine causes the death of the myelinating cells by triggering cell signaling pathways that induce oxidative stress, mitochondrial dysfunction, apoptosis, inflammation, endothelial/vascular dysfunctions, and neuronal and axonal damage [213,214,215,216,217].

To date, there is no cure. The only disease-modifying treatment currently available is hematopoietic stem cell transplantation, which is effective only when performed before symptoms appear, while other treatment options are symptomatic [218].

Gene therapy-based clinical trials (NCT04693598 and NCT05739643) are active; no clinical trials for the evaluation of dietary intervention exist. Through searches about diet and nutrient intake in Krabbe disease, the biomedical databases retrieved up to 18 results, but only 4 results really fitted within the aim of this work. Preclinical studies performed in the twitcher mouse (the naturally occurring animal model of the disease) suggested (i) a galactose-free diet enriched in soy isoflavones and antioxidants (coenzyme Q10, glutathione, and isoflavonoids) [219] and supplementation with vitamin D3 [220] for delaying the onset of symptoms. Recently, dietary supplementation with n-3 polyunsaturated fatty acids led to a slowing of the phenotypic presentation of the disease and restoration of lipid mediator production [221], as oxidative stress induces an increase in isoprostanoids levels in mouse brains [222]. In cellular systems, it has been reported that the 3′,4′,7-trihydroxyisoflavone holds a GALC-addressed chaperoning activity [223,224], useful to increase residual enzymatic activity in fibroblasts from Krabbe patients.

Although there are little data in the literature, such scientific evidence conducted on *in vivo* models gives a possible guarantee that a correct supplement of foods containing antioxidant nutrients can be a potential treatment to improve the quality of life in Krabbe syndrome (Table 2).

#### 5.2.2. Pelizaeus–Merzbacher Disease

Pelizaeus–Merzbacher disease (OMIM # 312080) is an X-linked leukodystrophy characterized by developmental delay, nystagmus, hypotonia, spasticity, and variable intellectual deficit [225]. The disorder is due to mutations or dosage alterations of the proteolipid protein 1 (PLP1) gene, located at chromosome Xq22.2 [226,227].

Treatments are symptomatic, including drugs for seizures, and spasticity.

Non-clinical trials matched the query as nutritional intervention. Biomedical databases returned to our search six results, but only three of these really fitted within the aim of this work. It has been shown in a mouse model of disease that cholesterol supplementation can enhance myelination [228], but dietary cholesterol was ineffective in myelination of patients [229]. The authors explain this phenomenon by suggesting that mice have a disturbance of blood–brain barrier (BBB) integrity that allows access of cholesterol from the circulation into the brain. Moreover, the ketogenic diet seemed to improve myelination and axonal damage in a mouse model with preserved BBB integrity [229], while the supplementation of curcumin has been reported as an antioxidant therapy in a mouse model of disease-carrying additional copies of PLP1 [230].

#### 5.2.3. Miscellaneous

As reported above, hypomyelinating conditions have a general phenotypic description and genetic heterogeneity in common. In this context, it is not surprising that the same nutritional intervention has been proven as effective for multiple conditions.

It is the case of the ketogenic diet that was reported as beneficial for seizure control in patients with hypomyelinating leukodystrophy-14 (OMIM # 617899) and drug-resistant seizures [231] and for the alleviation of psychomotor regression in KARS-related mitochondrial dysfunction and progressive leukodystrophy (OMIM # 619147) [232].

### 5.3. Rare Genetic Epilepsy

Developmental and epileptic encephalopathies represent a clinically and genetically heterogeneous group of age-dependent neurologic disorders characterized by the onset of refractory seizures in infancy or early childhood. Several disorders fitting into this group, such as the West syndrome (OMIM # 308350) with a common genetic mutation in the ARX gene (Ch. Xp21.3) [233], the Ohtahara syndrome [234], the Dravet syndrome (OMIM # 607208) with a common genetic mutation in the *SCN1A* gene (Ch. 2q24.3) [235], and the Lennox–Gastaut syndrome [236], have an often unclear etiology. Recently, the term “Infantile Spasm Syndrome” has been applied to include both West syndrome as well as conditions characterized by epileptic spasms in children even in the absence of all inclusion diagnostic criteria for West syndrome [237]. As recently reported by Ramantani et al. [238], the main proposed nutritional intervention for these conditions—that are often refractory to the antiepileptic drugs [234]—is the ketogenic diet [239,240].

Up to now, we found four clinical studies based on diet for West syndrome/infantile spasm (NCT01006811; NCT05279118; NCT01549288; and NCT00968136). Among these (i) the trial about the modified Atkins diet in infantile spasms refractory to hormonal therapy was withdrawn (NCT01549288); (ii) the trial NCT05279118 is active, but not recruiting, and it aims to compare the effect of the ketogenic diet to those of Adrenocorticotropic hormone (selected as active comparator); and (iii) two trials have been completed (NCT00968136, and NCT01006811) without published results on clinicaltrial.org. However, searching the clinical trials’ ID on the public database, the results of NCT01006811 were found and reported a 40% rate of spasm resolution after three months of modified Atkins diet (a less restrictive ketogenic diet) in children [241].

At the molecular level, it seems that the efficacy of the ketogenic diet could be strongly related to genetic variants, and consequently to rare disorders. Ko et al. [242] reported that patients identified as responders to the ketogenic diet were 77.8% in the case of Dravet syndrome; 35.8% in the case of West syndrome; 29.1% for the Lennox–Gastaut syndrome; and 64.7% in the case of Ohtahara syndrome. It must be pointed out that the nutritional intervention seemed to be effective in patients with SCN2A, STXBP1, KCNQ2, and SCN1A mutations that reported a responder rate of 100, 100, 83.3, and 77.8%, respectively. However, it was not effective in patients with CDKL5 mutations (responder rate = 0.0%) 3 months after implementation [86,242]. 

The reported results strongly suggest that the application of technologies, such as next-generation sequencing [243], could improve the understanding of the pathophysiology of each genetic mutation, enhancing the development of precision medicine to identify in which patients the ketogenic diet can ensure an efficient outcome.

The effectiveness of the ketogenic diet in the control of patients’ seizures has been also supported by preclinical studies in animal models, as reviewed by Griffin et al. [244] and demonstrated in mouse models for Dravet syndrome [245,246,247], and for West syndrome [248].

Recently, a trial has been approved on melatonin supplementation in the treatment of infantile spasms (https://trialsearch.who.int/Trial2.aspx?TrialID=ChiCTR2000036208 accessed on 6 September 2024).

Among the neuro-nutraceutical substances, cannabinoids or derivatives of *Cannabis sativa* have been largely investigated to treat rare epilepsy. The diet supplementation with cannabidiol-enriched extract has been reported to reduce seizures both in humans [249,250] and in animal models [251]. Specifically, cannabidiol oral solution was efficacious for the treatment of patients with drop seizures associated with Lennox–Gastaut syndrome (NCT02224690) [252], or with Dravet syndrome (NCT02091375) [253]. According to these results, cannabidiol (Epidyolex) received the orphan drug designation by the U.S. Food and Drug Administration and by the European Medicines Agency.

### 5.4. Rare Forms of Ataxia

Ataxia is a disease characterized by impaired muscle control during voluntary movements such as walking or grasping objects. It is usually the consequence of damage to the cerebellum, the brain structure responsible for muscle coordination. The etiology varies from sporadic, hereditary, and acquired forms.

Among the rare forms of ataxia, Friedreich ataxia, Stiff Person Syndrome, and Gluten ataxia benefit from nutritional intervention.

Friedreich ataxia is the most common of the inherited ataxias, caused mainly by homozygous GAA triplet repeat expansion in intron 1 of the FXN gene encoding for frataxin [254], a mitochondrial protein involved in iron metabolism. The supplementation with antioxidant agents, like vitamin E plus coenzyme Q10, and vitamin B1 decreases oxidative stress and enhances mitochondrial function [255]. Moreover, the antioxidant resveratrol increases FXN gene expression in cell models [256], and it seems useful in patients [257]. Recently, the clinical trial “Micronised resveratrol as a treatment for Friedreich Ataxia” (NCT03933163) has been completed, but no result was shared. Actually, there is a recruiting trial that will investigate the effect of combined physical exercise and dietary supplementation with Nicotinamide riboside (as NAD+ precursor) on skeletal muscle mitochondrial oxidative phosphorylation capacity, muscle mass, aerobic capacity, and glucose homeostasis (NCT04192136).

Stiff person syndrome is a rare autoimmune disease associated with cerebellar ataxia. Most patients have high levels of glutamic acid decarboxylase (GAD) antibodies. The etiology remains unclear. Gluten ataxia is often associated with Spinocerebellar ataxia type 35 with mutation in the Transglutaminases (TGM6) gene or autoantibodies against TGM6. In both diseases, a gluten-free diet has demonstrated good clinical effectiveness [258,259] (NCT00006492).

For the symptomatic treatment, some edible mushrooms have been suggested for their neuroprotective properties. *Pleurotus giganteus*, *Ganoderma lucidium*, and *Hericium erinaceus* contain bioactive compounds such as terpenoids, polysaccharides, alkaloids, and antibiotics that induce both the activation of the master regulator of the antioxidant defenses and the synthesis of Nerve Growth Factor, determining a slowing down of neuronal senescence [260].

Still under discussion today are the endocannabinoids that have been extensively studied in diseases associated with nerve cell damage, demonstrating that endocannabinoids show a broad spectrum of neuroprotective activities (antioxidant, anti-inflammatory, and pro-neurogenic) [261]. The endocannabinoid system is under investigation in spinocerebellar Ataxia Type-3 and other autosomal-dominant cerebellar ataxias [262,263]. As summarized by Gómez-Ruiz et al. [262], elevated levels of cannabinoid type-2 and cannabinoid type-1 receptors were found in spinocerebellar ataxias animal models and patients’ brain tissue. Specifically, in *post mortem* tissue of patients, the expression of cannabinoid type-2 receptors was found to be elevated in surviving neuronal cerebellar subpopulations (e.g., Purkinje cells, neurons of the dentate nucleus) [264] in which these receptors usually have a low physiological expression. The cannabinoid type-1 receptors were elevated in several cerebellar cells both in mouse models [265] and in *post mortem* patient tissue [264]. Globally, the collected evidence suggests that the alterations in endocannabinoid receptors may be related to the pathogenesis of ataxia and may be therapeutic targets. Results from *in vivo* studies suggested the potential role of cannabinoid type-1 receptor modulation for the management of ataxia symptoms in a rat model, but results were not fully satisfying, probably because the receptor may not act solely and other receptors should be considered [266].

### 5.5. Rare Brain Tumors

Based on the guidelines of the World Health Organization (WHO), rare brain tumors are classified according to several criteria, first and foremost, the histotype. To support the classification, several parameters are added such as morphological characteristics, growth characteristics, and ultimately, the molecular pattern involved in the genesis and development of the tumor [267,268]. As reported by Louis et al. [267], new tumor types and subtypes have been classified thanks to novel diagnostic technologies such as DNA methylome profiling.

Among the rare brain tumors, the most common malignant tumors that affect children are embryonal tumors (Medulloblastoma) [269], Pineal Region Tumors (Pineoblastoma) [270], Choroid Plexus Carcinoma [271], and Glial Tumors [272].

The nutritional factors on glioma incidence have been recently reviewed by Bielecka and Zukowsaka [273], suggesting a balanced diet containing fruits, vegetables, antioxidant-rich foods, omega-3 fatty acids, and adequate protein. As for the effect of dietary antioxidant vitamin intake on glioma risk in humans, published data seem to be controversial: some reports sustained their protective role [274,275], while other researchers described an apparent positive or null influence [276,277].

A recent systematic review and meta-analysis synthesized the associations between dietary antioxidant vitamin intake and risk of glioma by reviewing the available data up to March 2024 [278]. Even if the reading of this work is recommended [278], the major findings were that a (i) high intake of vitamin C was significantly associated with a lower risk of glioma and that (ii) high intake of vitamin A and vitamin E were not associated with the risk of glioma. In line with these points in a meta-analysis of 15 studies (including 13 control cases) assessing the impact of vitamin C intake on the risk of glioma, a higher intake of vitamin C than lower consumption was significantly related to a lower risk [279], while in contrast, previous studies reported that dietary vitamin A intake could reduce the glioma risk [280].

Thus, it seems clear that further clinical studies with detailed doses and large-scale prospective studies will be necessary, considering the encouraging results from preclinical analyses. To date, it was shown that (i) vitamin A (all-trans retinoic acid) stimulated apoptosis and had an inhibitory effect on the migration, invasion, and proliferation of glioma cell lines (U87 and SHG44) [281] and of primary cultures established from biopsies [282] and that (ii) the lipophilic derivative of vitamin C (ascorbile stearate) had a proapoptotic and antiproliferative effect in human glioma cells [283].

The ketogenic diet has been reported as useful for gliomas (including glioblastomas) by pieces of evidence from *in vitro*, *in vivo*, and clinical observations [284,285]. The rationale behind using a ketogenic diet for gliomas started from the observation that cancer cells, including glioma cells, often rely heavily on glucose for energy through the process of glycolysis (the “Warburg effect”) [286]. By restricting glucose availability, the ketogenic diet may impair the energy supply to tumor cells while providing an alternative energy source (ketones) to normal brain cells. Preclinical studies reported that the ketogenic diet not only enhanced survival and slowed tumor growth, but it also potentiated the effect of radiation by extending the survival of a mouse model of malignant glioma [287]. From a mechanistic point of view, the improved survivability seemed possibly due to a reduction in reactive oxygen species levels, and a modulation—toward the physiological levels—of gene expression involved in oxidative stress and antioxidant defense pathways [288]. In recent years, numerous clinical studies have been conducted as reported in Table 3. The data from a recent metanalysis pointed out a global positive effect of the ketogenic diet on patient survival as an adjuvant therapy of malignant gliomas [289].

The Li–Fraumeni syndrome is a rare genetic condition, also acting as a predisposition to cancer development in the brain. It is caused by point mutations in the p53 gene (TP53 on chromosome 17p13), which phenotypically cause uncontrolled cell proliferation [295]. Dietary supplementation with Nicotinamide riboside—a vitamin B3 dietary supplement—is under investigation (ClinicalTrials.gov ID NCT03789175).

Ongoing pieces of research are investigating the direct link between maternal diet during pregnancy and the development of rare brain tumors in offspring [296,297,298], suggesting that (i) foods generally associated with increased risk were cured meats, eggs/dairy, and oil products; foods generally associated with decreased risk were yellow-orange vegetables, fresh fish, and grains; (ii) maternal passive smoking and consumption of caffeinated beverages during pregnancy should be considered as a risk for glioma; (iii) oil products increased the risk of medulloblastoma; and (iv) diets higher in fruit and lower in fried foods and cured meats during pregnancy may reduce the risk of unilateral retinoblastoma.

## 6. Conclusions

The relationship between nutritional intervention and rare neurological diseases is an emerging area of research that may help to manage symptoms, slow disease progression, or even impact the underlying mechanisms of certain rare neurological disorders. The complex interactions between diet and neurological health require closer interdisciplinary collaboration between neurologists, dietitians, and researchers. It is crucial to conduct clinical trials to establish evidence-based dietary guidelines tailored to these unique conditions.

The major limits of this study reside in (i) the limited availability of data as accessed by the reduced number of published papers and completed clinical trials; (ii) the wide variety of the disease’s pathophysiology; and (iii) the underestimated influence of individual variability due to genetic differences. Specifically, as reported above, the number of published articles about the role of diet/nutrients on rare neurological disorders is very small. Clinical data often derive from case reports or small cohort studies, and large-scale clinical trials are scarce due to the rarity of these disorders and the huge heterogeneity of clinical manifestations. Moreover, even if preclinical studies give us more detailed information about the impact of a specific nutrient on cell function or on disease progression in animal models, they could not reflect the patients’ individual genetic differences and their global health status (e.g., integrity of anatomic structure and physiological functions as blood circulation, pressure, constipation, and deglutition). These latter aspects may influence how patients assume, metabolize, and distribute nutrients as has been discussed above in the case of cholesterol supplementation for Pelizaeus–Merzbacher disease.

Moreover, when searching for dietary recommendations for a specific rare disease, a great limit is the heterogeneity of occurring genetic mutations that turn into different phenotypes and clinical manifestations. Briefly, what could work for one patient may not be effective for another one, as enlightened here by reviewing the activity of trihydroxyisoflavone as a GALC-addressed chaperone.

However, despite these limitations, our analysis has, as a strength, the identification of a specific dietary intervention for the management of some symptoms. It is the case of ketogenic diet efficacy in different rare neurological diseases that share epilepsy in clinical manifestation. This finding may drive new trials for the evaluation of the ketogenic diet in epileptic patients with different rare disorders. It should be useful in order to identify if a putative leitmotif of treatment is possible and if the diet interacts with the specific pharmacological medications already in use. Moreover, the evidence reviewed here may also open the suggestion of a more detailed nutritional plan during pregnancy.

Globally, our work suggests that the field of rare diseases is a “societal laboratory” that is predicting future trends in patient-centered human healthcare, and as such, it is a model for personalized medicine in some aspects. Personalized nutrition, which considers the individual’s genetics, the microbiome, and the metabolic profile, is likely to play a pivotal role in future therapies. With advancements in technology, such as metabolomics and neuroimaging, it may be possible to identify biomarkers that predict dietary responses. According to these observations, the main challenge for the future may be to tailor nutrition and diet based on the individual’s specific genetic and biochemical profile. It may be difficult, especially in the case of rare neurological diseases, because the approach may be expensive, difficult for the caregivers, and not available in all healthcare settings due to the required involvement of specialized healthcare professionals and sophisticated tools.

This work critically describes the actual scenario, hoping to inspire future investigations for a deeper understanding of these diseases and to address the actual knowledge gaps. As described, in many cases, the mechanisms by which specific nutrients or diets impact neurological health are poorly understood. The lack of mechanistic understanding hampers the development of more targeted nutritional therapies, as in the case of the ketogenic diet and cannabinoid supplementation. Moreover, we found that the majority of studies focus on short-term outcomes (seizure reduction, myelinization, or symptom improvements), but the long-term effects are often unknown and unexplored.

Taken together, in some rare neurological diseases, diet may become an integral part of holistic care strategies, and it needs to receive the same attention as pharmacological treatments, offering patients not only improved quality of life but also potentially new avenues for disease prevention, management, and treatment.

## Figures and Tables

**Figure 1 nutrients-16-03114-f001:**
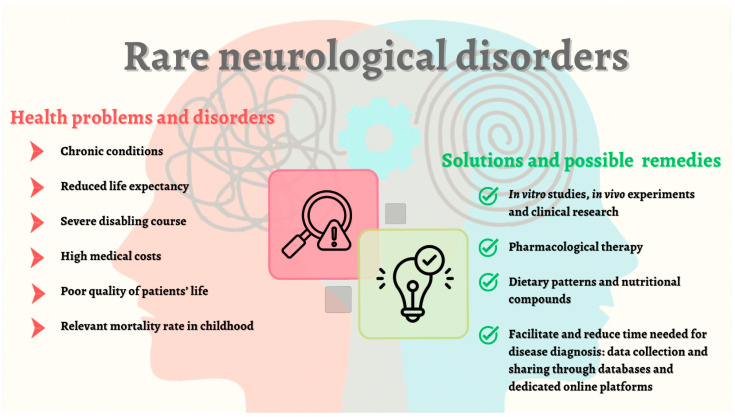
Main characteristics of rare neurological diseases and possible investigations for symptomatic treatments (created with BioRender.com, accessed on 3 September 2024).

**Figure 2 nutrients-16-03114-f002:**
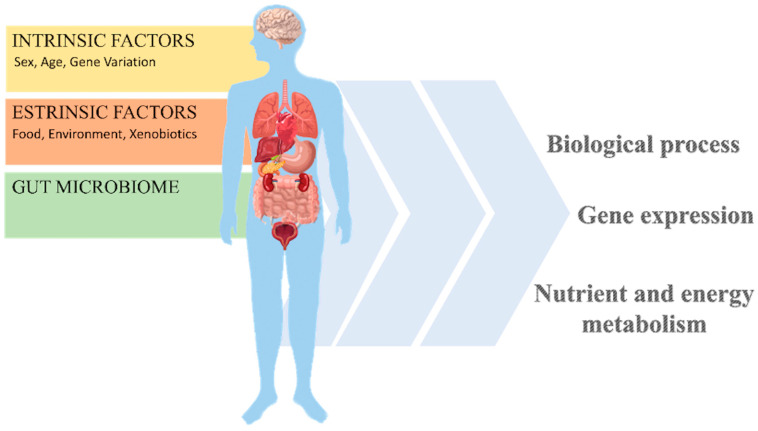
Factors that influence alimentary wellness by modulator effects on the main homeostatic functions (created with BioRender.com, accessed on 27 June 2024).

**Figure 3 nutrients-16-03114-f003:**
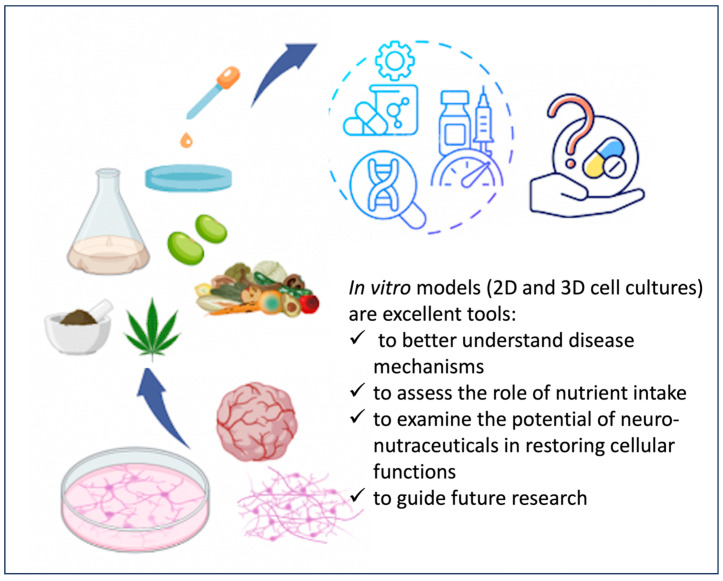
*In vitro* models as tools in rare neurological disease research and therapeutic development (partially created with BioRender.com, accessed on 7 August 2024).

**Figure 4 nutrients-16-03114-f004:**
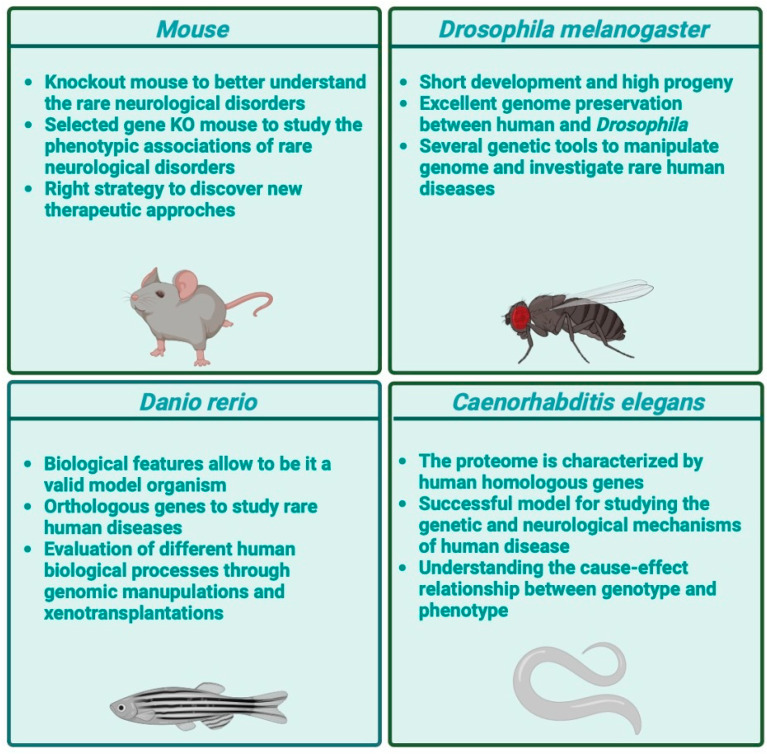
Animal models for rare neurological diseases (created with BioRender.com, accessed on 3 September 2024).

**Figure 5 nutrients-16-03114-f005:**
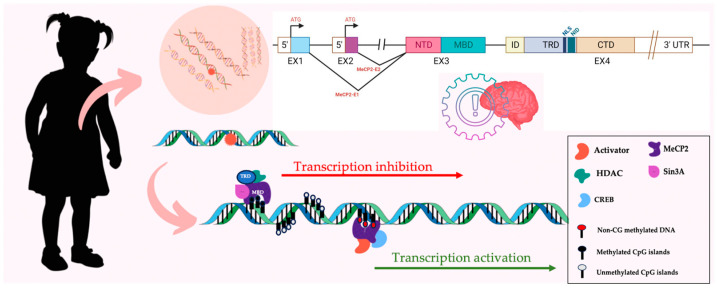
MeCP2 gene structure and its activity on target genes. The meCP2 gene has N-terminal (NTD); methyl binding (MBD); intervening (ID); transcription repression (TRD); and C-terminal (CTD) domains. MeCP2 recruits a transcriptional corepressor complex containing Sin3A and histone deacetylase (HDAC) to methylated CpG islands and induces transcription inhibition in the target gene (TRD, transcriptional repression domain; MBD, methyl-CpG-binding domain). MeCP2 can activate gene transcription by recruiting CREB and other transcriptional factors to non-methylated CG DNA regions (partially created with BioRender.com, accessed on 10 August 2024).

**Table 2 nutrients-16-03114-t002:** Nutritional interventions for Krabbe disease.

Nutritional Interventions Krabbe Disease		
	**Nutrients or Neuro-Nutraceutical Substance**	**Effects**
**Nutrient supplementation**	Vitamin D3 ^1^	Delay symptoms onset [220]
	n-3 polyunsaturated fatty acids ^1^	Slow the phenotypic presentation [221]
	Soy isoflavones and antioxidants (coenzyme Q10, glutathione and isoflavonoids ^1^	Delay symptoms onset [219]
	3′,4′,7-Trihydroxyisoflavone ^2^	Pharmacological chaperone [223,224]
	**Type of diet**	**Effects**
**Dietary modification**	Galactose-free diet ^1^	Delay symptoms onset [219]

^1^ Tested in animal models. ^2^ Tested in *in vitro* model.

**Table 3 nutrients-16-03114-t003:** Clinical trials for the ketogenic diet in gliomas.

Clinical Trials for Ketogenic Diet in Gliomas		
Status	Clinical Trial ID/Name	Results
Withdrawn	NCT05373381 The KetoGlioma (Ketogenic Glioma) Study	Study is on hold indefinitely due to funding issues
Unknown status	NCT03278249 Feasibility Study of Modified Atkins Ketogenic Diet in the Treatment of Newly Diagnosed Malignant Glioma	
	NCT02939378 Ketogenic Diet Adjunctive to Salvage Chemotherapy for Recurrent Glioblastoma: a Pilot Study	
Terminated	NCT02046187 Ketogenic Diet with Radiation and Chemotherapy for Newly Diagnosed Glioblastoma	Excessive protocol deviations due to strict nature of diet requirements
Completed	NCT02286167 Glioma Modified Atkins-based Diet in Patients with Glioblastoma	Production of ketonuria and significant systemic and cerebral metabolic changes in participants [290]
	NCT03075514 Ketogenic Diets as an Adjuvant Therapy in Glioblastoma	No posted results, only a linked paper with trial description [291]
	NCT02302235 Ketogenic Diet Treatment Adjunctive to Radiation and Chemotherapy in Glioblastoma Multiforme: a Pilot Study NCT01865162 Ketogenic Diet as Adjunctive Treatment in Refractory/End-stage Glioblastoma Multiforme: a Pilot Study (KGDinGBM)	Diet was well tolerated. The small sample size limits efficacy conclusions [292].
	NCT00575146 Ketogenic Diet for Recurrent Glioblastoma	Diet was feasible and safe but probably has no significant clinical activity when used as single agent in recurrent glioma [293].
	NCT01754350 Calorie-restricted, Ketogenic Diet and Transient Fasting During Reirradiation for Patients With Recurrent Glioblastoma	Diet was feasible and effective in inducing ketosis in heavily pretreated patients with recurrent glioma, but failed to increase the efficacy of reirradiation [294].
Recruiting	NCT05708352 A Phase 2 Study of the Ketogenic Diet vs. Standard Anti-cancer Diet Guidance for Patients With Glioblastoma in Combination With Standard-of-care Treatment	
	NCT04691960 A Pilot Study of Ketogenic Diet and Metformin in Glioblastoma: Feasibility and Metabolic Imaging	
	NCT05564949 A Ketogenic Diet as a Complementary Treatment on Patients With High-grade Gliomas and Brain Metastases	
	NCT05183204 Paxalisib With a High Fat, Low Carb Diet and Metformin for Glioblastoma	
	NCT04730869 Metabolic Therapy Program In Conjunction With Standard Treatment For Glioblastoma	
Active, not recruiting	NCT03451799 Ketogenic Diet in Combination With Standard-of-care Radiation and Temozolomide for Patients With Glioblastoma	
	NCT01535911 Pilot Study of a Metabolic Nutritional Therapy for the Management of Primary Brain Tumors	

## Data Availability

Data sharing is not applicable.

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
