# Peer review of "Diet and Nutrients in Rare Neurological Disorders: Biological, Biochemical, and Pathophysiological Evidence"

_nutrients, 2024, doi:10.3390/nu16183114_

Round 1

Reviewer 1 Report

Comments and Suggestions for Authors

The paper is a traditional review concerning rare neurological disorders and the effect of diet and nutrients a key factor discussed in this work. Introduction justifies the review, however authors, citing a respected work published in the Lancet, indicate the alarming prevalence of nervous system disorders, but for the sake of honesty it would be worth pointing out that these alarming statistics in case of 93% concern tension-type headache and migraine (3.17 out of 3.4 bilion cases). The work was written in a logical and consistent manner, subsequent chapters introduce the reader to the subject, and at the end move on to specific examples of how nutritional interventions and nutrients can alleviate the symptoms of the diseases discussed. And here I feel a certain dissatisfaction, I think that in several cases more details could be provided, especially in case of:

- Rare genetic epilepsy (line 686-689) - what do you mean by writing “the ketogenic diet […] seems to be particularly effective”?

- Rare forms of ataxia - what do you mean by writing “Experimental data show a promising activity of cannabinoids”?

- Rare brain tumors - writing “they reported limited but positive evidence of the role of vitamin A and C in decreasing the glioma risk” give more details, and in case of “ketogenic diet has been reported useful for gliomas and glioblastomas” give some numbers.

The last sentence in the conclusion part is not entirely true because in some cases diet is not effective.  It would be nice in the conclusions to give some names of the diseases and examples of the successful nutrients or diets application. This will increase the chances of citing of your work because we can assume that not all readers will have time to read all 18 pages of your review.

Besides:

Use italics for all microbiological names, e.g. Cannabis sativa, line 277.

There is no need to use capital letters in case of names of some vitamins and minerals

Author Response

Response to Reviewer 1 Comments

1. Summary

Thank you very much for taking the time to review our manuscript. Please find the detailed responses below. The corresponding revisions/corrections are written in red in the re-submitted files.

2. Questions for General Evaluation

Reviewer’s Evaluation

Response and Revisions

Is the work a significant contribution to the field?

4 stars

Thank you.

We really appreciate your feedback and we hope you enjoy the improvement we made based on your suggestions, as detailed in the point-by-point response.

Is the work well organized and comprehensively described?

4 stars

Is the work scientifically sound and not misleading?

4 stars

Are there appropriate and adequate references to related and previous work? 

5 stars

Is the English used correct and readable?        

5 stars

3. Point-by-point response to Comments and Suggestions for Authors

Comments 1: The paper is a traditional review concerning rare neurological disorders and the effect of diet and nutrients a key factor discussed in this work. Introduction justifies the review, however authors, citing a respected work published in the Lancet, indicate the alarming prevalence of nervous system disorders, but for the sake of honesty it would be worth pointing out that these alarming statistics in case of 93% concern tension-type headache and migraine (3.17 out of 3.4 bilion cases). The work was written in a logical and consistent manner, subsequent chapters introduce the reader to the subject, and at the end move on to specific examples of how nutritional interventions and nutrients can alleviate the symptoms of the diseases discussed.

Response 1: Thank you for this comment. It is very important for us that you recognized our work was written in a logical and consistent manner. We thank for the comment about the respected work published in the Lancet. We agree with this comment. Therefore, we have added some data in order to pointed out the findings, and the relevance of the cited paper in our background. In the revised manuscript this change can be found – page 1, paragraph 1, lines 39-42.

Comments 2: And here I feel a certain dissatisfaction, I think that in several cases more details could be provided, especially in case of: Rare genetic epilepsy (line 686-689) - what do you mean by writing “the ketogenic diet […] seems to be particularly effective”?

Response 2: Accepted and done. Results has been detailed, as requested. Thank you. We think that our paper is now more informative and we hope to have satisfied your observation (lines 742-779).  

Comments 3: - Rare forms of ataxia - what do you mean by writing “Experimental data show a promising activity of cannabinoids”?

Response 3: Thank for you comment. We wrote about the alterations in endocannabinoid receptors and results from in vivo studies were discussed (lines 817-830).

Comments 4: - Rare brain tumors - writing “they reported limited but positive evidence of the role of vitamin A and C in decreasing the glioma risk” give more details,

Comments 5: and in case of “ketogenic diet has been reported useful for gliomas and glioblastomas” give some numbers.

Response 4 and response 5: Globally, we agree with both the comments. Thank you. We wrote about the controversial data regarding the role of dietary vitamins intake on glioma risk and on cell fate (lines 847-866).

Similarly, for the ketogenic diet on gliomas and glioblastomas, more details and a table have been added (lines 867-883).

Comments 6: The last sentence in the conclusion part is not entirely true because in some cases diet is not effective.  It would be nice in the conclusions to give some names of the diseases and examples of the successful nutrients or diets application. This will increase the chances of citing of your work because we can assume that not all readers will have time to read all 18 pages of your review.

Response 6: Agree. We have, accordingly, modified the last sentence of the conclusion. Moreover, according to Your suggestion, we added (in a brief and read-friendly way) some examples of conditions that drive us to define the major limits of our work, the main future challenge (with putative limits), and the principal gaps of knowledge we identified and we hope to be addressed.

Comments 7: Besides:

Use italics for all microbiological names, e.g. Cannabis sativa, line 277.

There is no need to use capital letters in case of names of some vitamins and minerals

Response 7: Agree. Accepted and done.

Reviewer 2 Report

Comments and Suggestions for Authors

This innovative review summarizes the current knowledge on the proposed and performed dietary interventions in rare neurological disorders. This information is preceded by a broad background inroducing rare neurological disorders and methodology of their studies. The information on dietary interventions is modest with respect to this background and apparently no simple synthesis is possible given the heterogeneity of causes and metabolic mechanisms of these diseases. Nevertheless, the review critically describes the actual state of affairs and may be an inspiration for  future studies, as pointed by the authors in Conclusions.

Remarks:

Figure 1, „Solutions and possible remedies”: shouldn’t in vitro studies precede in vivo experiments?

Lines 197/198: „direct modifiers of protein function, powerful protein molecules”, this statement could be re-phrased in a more logical form

Line 257:”usefully”or „useful”?

Lines 277/278 and 282: „Cannabis  sativa”, please in italics

Figure 4: „High homology with the human genome” concerns all model organisms; stressing this fact only for C. elegans can make an impression that C. elegans genome has higher komolohy with the human genome than e.g. mouse.

Lines 442, 446, 450,452: „C. elegans”, please in italics

Lines 599/600: „antioxidant ω-3 polyunsaturated fatty acids”, are polyunsaturated fatty acids antioxidants? They are rather preferential substrates for peroxidation; their indirect antioxidant effects are consequences of the anti-inflammatory action

Comments on the Quality of English Language

Minor grammar errors; spelling should be uniformized

Author Response

Response to Reviewer 2 Comments

1. Summary

Thank you very much for taking the time to review this manuscript. Please find the detailed responses below. The corresponding revisions/corrections are written in red in the re-submitted files.  

2. Questions for General Evaluation

Reviewer’s Evaluation

Response and Revisions

Is the work a significant contribution to the field?

5 stars

Thank you.

We really appreciate your feedback and we hope you enjoy the improvement we made based on your suggestions, as detailed in the point-by-point response.

Is the work well organized and comprehensively described?

4 stars

Is the work scientifically sound and not misleading?

4 stars

Are there appropriate and adequate references to related and previous work? 

5 stars

Is the English used correct and readable?        

4 stars

3. Point-by-point response to Comments and Suggestions for Authors

Comments 1: This innovative review summarizes the current knowledge on the proposed and performed dietary interventions in rare neurological disorders. This information is preceded by a broad background inroducing rare neurological disorders and methodology of their studies. The information on dietary interventions is modest with respect to this background and apparently no simple synthesis is possible given the heterogeneity of causes and metabolic mechanisms of these diseases. Nevertheless, the review critically describes the actual state of affairs and may be an inspiration for  future studies, as pointed by the authors in Conclusions.

Response 1: Thank you for these comments that inspired us to ameliorate the description of our work in the introduction. According to your comment, we added “ Therefore, in this narrative review, we first introduce readers to the topic of rare diseases by focusing on rare neurological disorders. Then, we analyzed the concept of "nutritional approaches" in neurological illness, trying to summarize the main methodologies applied for the study of nutritional interventions in rare neurological disorders with a critical evaluation of limits and strengths of in vitro and in vivo models, and of clinical studies. These preliminary indications supported the last part of the work in which we analyzed the scientific literature by reviewing the in vitro, in vivo and clinical evidence on the effects of diet and of nutrient intake on some rare neurological disorders, including rare developmental disorders (Angelman Syndrome, and Rett Syndrome), rare leukodystrophies (Krabbe disease, Pelizaeus–Merzbacher disease), rare genetic epilepsy, rare forms of ataxia, and rare brain tumors. Specifically, we review the evidence available on the following web-based databases: PubMed, Web of Science Core Collection, and DynaMed. Moreover, for clinical trials we searched on clinicaltrials.gov. Globally, to focus our search, we applied the Boolean operator “AND” to combine the condition/disease with specific terms (diet, or nutrition, or nutrients). When a specific term turned results in line with the aims of this review, closer research was done by the inclusion of the identified keywork. (lines 97-113). Here, in italic, the changes made.

Comments 2: Remarks: Figure 1, „Solutions and possible remedies”: shouldn’t in vitro studies precede in vivo experiments?

Response 2: Thank you. We agree with your note. The change you suggested has been made in figure 1.

Comments 3: Lines 197/198: „direct modifiers of protein function, powerful protein molecules”, this statement could be re-phrased in a more logical form

Response 3: Thank you. The statement has been removed because the concept is redundant.

Comments 4: Line 257:”usefully”or „useful”?

Response 4: Thank you. Correction made “useful” (line 277) .

Comments 5: Lines 277/278 and 282: „Cannabis  sativa”, please in italics

Response 5: Thank you. Accepted and done. Cannabis sativa has been rewritten in italics as suggested (lines 297/298 and 302).

Comments 6: Figure 4: „High homology with the human genome” concerns all model organisms; stressing this fact only for C. elegans can make an impression that C. elegans genome has higher komolohy with the human genome than e.g. mouse.

Response 6: Thank you. Accepted and done. The revised figure has been added.

Comments 7: Lines 442, 446, 450,452: „C. elegans”, please in italics

Response 7: Thank you. Accepted and done.

Comments 8: Lines 599/600: „antioxidant ω-3 polyunsaturated fatty acids”, are polyunsaturated fatty acids antioxidants? They are rather preferential substrates for peroxidation; their indirect antioxidant effects are consequences of the anti-inflammatory action

Response 8: Agree. We have, accordingly, revised the sentences, by removing the word “antioxidant”.

4. Response to Comments on the Quality of English Language

Point 1: Minor grammar errors; spelling should be uniformized

Response 1: Thank you for pointing this out. We agree with this comment and a global revision has been done as you can see in the marked version of our work. Grammatical errors have been corrected and the spelling has been uniformized.

Reviewer 3 Report

Comments and Suggestions for Authors

This review explores the role of diet and nutrients in rare neurological disorders by evaluating the data in the literature about the biological, biochemical, and pathophysiological dimensions of this type of pathology. The topic is interesting due to the scarcity of therapeutic interventions for rare neurological diseases. Proofreading is needed to correct syntactical and grammatical errors. Please refer to the additional observations below:

Abstract- please consider structuring this section in a Background-Objective-Methods- Results-Conclusion(s) format because, in its current form, it is not clear to the readers what methodology was used, what the results are, and what is the Authors’ conclusion.

Lines 45-46- consider rephrasing that sentence for comprehensiveness;

Lines 63-65- the same as above;

Lines 90-99- How was the literature analyzed, more exactly? Is this a narrative review, a scoping review, or a systematic one? What databases were searched and what types of articles were included/excluded? In order to ensure the replicability of the results (and their validity, in consequence), more specific data on the methods used to explore the literature is needed.

Fig.1- revise „chronical debilitate clinical conditions”;

It is unclear why chapters 2 and 3 are not included in the „Introduction” section, because they do not include data referring to the objective of the review, formulated in lines 90-99.

Chapter 4—If this section is to be considered part of the „Results” section, then maybe add another objective about reviewing data on the clinical and preclinical models for testing nutritional interventions in rare neurological disorders.

Chapter 5- I assume this is where the results of the literature review begin because this section would correspond to the formulated objective. Please state how many primary and secondary sources the „Results” section is based on. Even if this is a narrative review and no quality-for-evidence methods were applied, it would be important to offer an image of the whole body of evidence collected before analyzing each dimension in detail. For example, how many preclinical and how many clinical trials were found on the subject? Any specifics of these studies/trials? Also, separating finalized trials/studies from those that are ongoing or planned to begin in the near future would be beneficial for the readers.

Lines 222-224- please revise that sentence;

Page 13- title 5.1.2 is repeated;

Fig.2- footonote-„MecCP2” is a typo;

The current review's limitations and strengths need to be highlighted, and directions for future research need to be better structured.

Comments on the Quality of English Language

Please thoroughly revise the entire manuscript (figures included) for syntactical and grammar errors.

Author Response

Response to Reviewer 3 Comments

1. Summary

Thank you very much for taking the time to review this manuscript. Please find the detailed responses below. The corresponding revisions/corrections are written in red in the re-submitted files.

2. Questions for General Evaluation

Reviewer’s Evaluation

Response and Revisions

Is the work a significant contribution to the field?

 2 stars

As reported in the following section “point-by-point response”, we tried to follow your observations and suggestions in order to improve the quality of our work.

Is the work well organized and comprehensively described?

1 star

Is the work scientifically sound and not misleading?

2 stars

Are there appropriate and adequate references to related and previous work? 

3 stars

Is the English used correct and readable?        

3 stars

3. Point-by-point response to Comments and Suggestions for Authors

Comments 1: This review explores the role of diet and nutrients in rare neurological disorders by evaluating the data in the literature about the biological, biochemical, and pathophysiological dimensions of this type of pathology. The topic is interesting due to the scarcity of therapeutic interventions for rare neurological diseases. Proofreading is needed to correct syntactical and grammatical errors.

Response1: Thank you for pointing this out. We agree with this comment and a global revision has been done as you can see in the marked version of our work.

Comments 2: Please refer to the additional observations below:

Abstract- please consider structuring this section in a Background-Objective-Methods- Results-Conclusion(s) format because, in its current form, it is not clear to the readers what methodology was used, what the results are, and what is the Authors’ conclusion.

Response 2: Thank you for pointing this out. Considering that it is a narrative review, the suggested structure of the abstract could be not applicable and misinformative, offering the possibility to confuse the readers. Moreover, the abstract has been preliminary evaluated and selected by editorial staff (May 2024). For these reasons, we prefer to maintain the same abstract.

Comments 3: Lines 45-46- consider rephrasing that sentence for comprehensiveness;

Response 3: Thank you. We agree. Accepted and done. The sentence is now “Although the etiology is heterogeneous, the NDs show systemic and long-term organic disabilities with high incidence of death, as supported by meta-analysis studies” (lines 51-53).

Comments 4: Lines 63-65- the same as above;

Response 4: Thank you. We agree. The sentence has been re-written: “Proper nutritional education could be a crucial intervention in NDs treatment to minimize the risks of malnutrition with likely repercussions on the worsening of symptoms related to neurological diseases [8]”, (lines 68-70).

Comments 5: Lines 90-99- How was the literature analyzed, more exactly? Is this a narrative review, a scoping review, or a systematic one? What databases were searched and what types of articles were included/excluded? In order to ensure the replicability of the results (and their validity, in consequence), more specific data on the methods used to explore the literature is needed.

Response 5: As reported above, and now written in revised manuscript (line 97), this is a narrative review and, according to your comment we added the following sentences: “Specifically, we review the evidence available on the following web-based databases: PubMed, Web of Science Core Collection, and DynaMed. Moreover, for clinical trials we searched on clinicaltrials.gov. Globally, to focus our search, we applied the Boolean operator “AND” to combine the condition/disease with specific terms (diet, or nutrition, or nutrients). When a specific term turned results in line with the aims of this review, closer research was done by the inclusion of the identified keywork” (lines 107-113).

Comments 6: Fig.1- revise „chronical debilitate clinical conditions”;

Response 6: Thank you. Accepted and done. The revised figure has been added.

Comments 7: It is unclear why chapters 2 and 3 are not included in the „Introduction” section, because they do not include data referring to the objective of the review, formulated in lines 90-99.

Comments 8: Chapter 4—If this section is to be considered part of the „Results” section, then maybe add another objective about reviewing data on the clinical and preclinical models for testing nutritional interventions in rare neurological disorders.

Response 7 and Response 8: We agree with both these comments that inspired us to ameliorate the description of our work in the introduction. According to your comment, we added “Therefore, in this narrative review, we first introduce readers to the topic of rare diseases by focusing on rare neurological disorders. Then, we analyzed the concept of "nutritional approaches" in neurological illness, trying to summarize the main methodologies applied for the study of nutritional interventions in rare neurological disorders with a critical evaluation of limits and strengths of in vitro and in vivo models, and of clinical studies. These preliminary indications supported the last part of the work in which we analyzed the scientific literature by reviewing the in vitro, in vivo and clinical evidence on the effects of diet and of nutrient intake on some rare neurological disorders, including rare developmental disorders (Angelman Syndrome, and Rett Syndrome), rare leukodystrophies (Krabbe disease, Pelizaeus–Merzbacher disease), rare genetic epilepsy, rare forms of ataxia, and rare brain tumors” (lines 97-107). Here, in italic, the changes made.

Comments 9: Chapter 5- I assume this is where the results of the literature review begin because this section would correspond to the formulated objective. Please state how many primary and secondary sources the „Results” section is based on. Even if this is a narrative review and no quality-for-evidence methods were applied, it would be important to offer an image of the whole body of evidence collected before analyzing each dimension in detail. For example, how many preclinical and how many clinical trials were found on the subject? Any specifics of these studies/trials? Also, separating finalized trials/studies from those that are ongoing or planned to begin in the near future would be beneficial for the readers.

Response 9: According to your suggestions, we have revised the main text to emphasize this point by adding the required information. Please, You can read:

-       lines 551-553 for Angelman Syndrome, specifically we added “As about diet and nutrients intake in Angelman Syndrome, we found two clinical trials and nineteen articles Among these latter, only thirteen deals with nutritional interventions”;

-       Lines 618-644 already reported data on the number of clinical trials and on preclinical researches in the field of diet and nutrition for Rett Syndrome;

-       Lines 682-685 for Krabbe disease: “Gene therapy-based clinical trials (NCT04693598 and NCT05739643) are active; none clinical trials for the evaluation of dietary intervention. Searching about diet and nutrients intake in Krabbe disease, the biomedical databases retrieved up to eighteen results, but only four results really fitted within the aim of this work.” Here, in italic, we show the added sentence.

-       Lines 709-711 for Pelizaeus–Merzbacher disease: “None clinical trials matched the query as nutritional intervention. Biomedical databases returned to our search six results, but only three really fitted within the aim of this work”.

The same has been done for the paragraphs: “Rare genetic epilepsy”, “Rare forms of ataxia”; and “Rare brain tumors”.

Comments 10: Lines 222-224- please revise that sentence;

Response 10:  The sentence has been revised “It is clear that the modulation of central nervous system homeostasis by nutrients could be both a risk factor for the development of neurological disease [50] as well as an opportunity to improve the health status by nutritional interventions” (line 241-243).

Comments 11: Page 13- title 5.1.2 is repeated;

Response 11: The title has been corrected.

Comments 12: Fig.2- footonote-„MecCP2” is a typo;

Response 12: Thank you. Accepted and done

Comments 13: The current review's limitations and strengths need to be highlighted, and directions for future research need to be better structured.

Response 13: Agree. We have, accordingly, modified the conclusion to emphasize this point. 

In a reader-friendly manner, we added the major limits of our work, the main future challenge (with putative limits), and we recapitulated the principal gaps of knowledge we identified and we hope to be addressed.

Comments on the Quality of English Language

Please thoroughly revise the entire manuscript (figures included) for syntactical and grammar errors.

Response:Accepted and done

Round 2

Reviewer 3 Report

Comments and Suggestions for Authors

The quality of the manuscript improved due to the diligent work of the authors.

It is still unclear how the reader could understand the methodology used, the review's results, and the Authors' conclusions from the current version of the "Abstract."

Author Response

Response to Reviewer 1 Comments

1. Summary

Thank you very much for taking the time to review our manuscript. Please find the detailed responses below. The corresponding revisions/corrections are written in red in the re-submitted files.

2. Questions for General Evaluation

Reviewer’s Evaluation

Response and Revisions

Is the work a significant contribution to the field?

3 stars

Thank you.

We really appreciate your feedback and we hope you appreciate the modification in the abstract, as detailed in the point-by-point response.

Is the work well organized and comprehensively described?

3 stars

Is the work scientifically sound and not misleading?

3 stars

Are there appropriate and adequate references to related and previous work? 

3 stars

Is the English used correct and readable?        

5 stars

3. Point-by-point response to Comments and Suggestions for Authors

Comments 1:  The quality of the manuscript improved due to the diligent work of the authors.

Response 1: Thank you for this comment. It is very important for us that you recognized the quality of our manuscript, and the work we made.

Comments 2:  It is still unclear how the reader could understand the methodology used, the review's results, and the Authors' conclusions from the current version of the "Abstract."

Response 2: Abstract has been modified with the introduction of some information. Thank you. We think that our abstract is now more informative and we hope to have satisfied your observation (lines 16-34).